# Radical in the Peroxide-Produced F-Type Ferryl Form of Bovine Cytochrome *c* Oxidase

**DOI:** 10.3390/ijms232012580

**Published:** 2022-10-20

**Authors:** Tereza Sztachova, Adriana Tomkova, Erik Cizmar, Daniel Jancura, Marian Fabian

**Affiliations:** 1Department of Biophysics, Faculty of Science, University of P. J. Safarik, Jesenna 5, 041 54 Kosice, Slovakia; 2Department of Condensed Matter Physics, Faculty of Science, University of P. J. Safarik, Park Angelinum 9, 040 01 Kosice, Slovakia; 3Center for Interdisciplinary Biosciences, Technology and Innovation Park, University of P. J. Safarik, Jesenna 5, 041 54 Kosice, Slovakia

**Keywords:** cytochrome oxidase, free radical, ferryl intermediate, electron paramagnetic resonance spectroscopy, isothermal titration calorimetry

## Abstract

The reduction of O_2_ in respiratory cytochrome *c* oxidases (CcO) is associated with the generation of the transmembrane proton gradient by two mechanisms. In one of them, the proton pumping, two different types of the ferryl intermediates of the catalytic heme *a*_3_-Cu_B_ center **P** and **F** forms, participate. Equivalent ferryl states can be also formed by the reaction of the oxidized CcO (**O**) with H_2_O_2_. Interestingly, in acidic solutions a single molecule of H_2_O_2_ can generate from the **O** an additional **F**-type ferryl form (**F^•^**) that should contain, in contrast to the catalytic **F** intermediate, a free radical at the heme *a*_3_-Cu_B_ center. In this work, the formation and the endogenous decay of both the ferryl iron of heme *a*_3_ and the radical in **F^•^** intermediate were examined by the combination of four experimental approaches, isothermal titration calorimetry, electron paramagnetic resonance, and electronic absorption spectroscopy together with the reduction of this form by the defined number of electrons. The results are consistent with the generation of radicals in **F^•^** form. However, the radical at the catalytic center is more rapidly quenched than the accompanying ferryl state of heme *a*_3_, very likely by the intrinsic oxidation of the enzyme itself.

## 1. Introduction

The final step of cellular respiration in aerobic organisms is the reduction of O_2_ to H_2_O catalyzed mainly by the membrane-bound heme-copper oxidases. Cytochrome *c* oxidases (CcO), the large subclass of heme-copper oxidases, are found in all mitochondria and some types of bacteria [1]. The reduction of oxygen in CcO is accomplished by electrons accepted from ferrocytochrome *c* (*c*^2+^). The electron transfer from *c*^2+^ to O_2_ is facilitated by four redox centers of CcO: Cu_A_, iron of heme *a* (Fe_a_), iron of heme *a*_3_ (Fe_a3_), and Cu_B_. A dinuclear copper center, Cu_A_, is the first acceptor of electrons from *c*^2+^. These electrons are rapidly distributed between Cu_A_ and Fe_a_ [2,3,4]. The intraprotein flow of electrons then continues from Fe_a_ to the catalytic binuclear Fe_a3_-Cu_B_ center where the reduction of O_2_ to H_2_O occurs (Figure 1). The catalytic center is also the site where the inhibitors (e.g., cyanide, azide, CO) of CcO can be bound.

The reduction of O_2_ in CcO proceeds via several distinct oxy-intermediates determined by the number of electrons and protons accepted by the catalytic Fe_a3_-Cu_B_ center (Figure 2) (for reviews [5,6,7,8,9,10]). From these intermediates, the most important and intriguing are the ferryl states that are divided, based on their optical spectra, into the **P** (**P_M_**, **P_R_**) and the **F** (**F**, **F^•^**)-type forms.

The **P_M_** is produced by the reaction of two-electron reduced (mixed-valence) CcO with O_2_ [11,12,13,14,15]. It was named **P_M_** since originally it was expected to have a peroxide bound at the catalytic center (e.g., Fe_a3_^3+^-O-O-Cu_B_^2+^). However, it has been shown later that the dioxygen bond in this intermediate is already cleaved by a four-electron reduction of O_2_. Two of them are coming from the oxidation of Fe_a3_^2+^ to Fe_a3_^4+^=O, one from the oxidation of Cu_B_^+^ to Cu_B_^2+^, and the donor of the fourth electron is plausibly the nearby Tyr244 (bovine CcO numbering), producing the free radical (YO**^•^**) at the vicinity of the catalytic center (Figure 1 and Figure 2).

Delivery of the third electron into the catalytic site of **P_M_** leads to the appearance of the second **P**-type intermediate, the **P_R_** (Figure 2). Since the radical is reduced in this intermediate and the visible absorption spectrum is almost identical to **P_M_** it was named **P_R_** [12,16,17,18,19]. A characteristic feature of these intermediates is a sharp α-band with a maximum at 607 nm in the difference spectra of **P_M_** and **P_R_** relative to the oxidized CcO.

One proton uptake into the catalytic center of the **P_R_** triggers the **P_R_**-to-**F** transition (Figure 2). This conversion is connected with the change in both the shape and the maximum of the α-band that is shifted from 607 to 580 nm [20,21,22,23,24,25,26,27,28]. With the transfer of an additional electron and proton into the **F** intermediate, the reduction of O_2_ to water is completed and the oxidized CcO (**O**) is recovered [20,21,22,23,24,25,26,29,30,31,32,33,34,35].

Analogous ferryl states can be produced by the reaction of the oxidized CcO with hydrogen peroxide as well [36,37,38,39,40,41]. Interestingly, the reaction of the **O** with peroxide can generate two types of the **F** form (**F**, **F^•^**). The peroxide-produced **F** state should be equivalent to the catalytic intermediate. Its formation by two molecules of H_2_O_2_ can be pictured by these two reactions:[Fe_a3_^3+^-OH HO-Cu_B_^2+^ YOH] + H_2_O_2_ → [Fe_a3_^4+^=O HO-Cu_B_^2+^ YO**^•^**] + 2H_2_O(1)
**O              P_M_**
[Fe_a3_^4+^=O HO-Cu_B_^2+^ YO**^•^**] + H_2_O_2_ → [Fe_a3_^4+^=O H_2_O-Cu_B_^2+^ YOH] + O_2_**^•−^**
(2)
**P_M_              F**
where the first molecule of H_2_O_2_ produces the **P_M_** form [37,39,42,43,44] and the second peroxide is expected to reduce the YO**^•^** and generate the **F** state and release the superoxide [38,45].

The conversion of the **P** form to the **F** state is not restricted only to the proton uptake. It has been proposed several times that in the transition from **P** to **F** form the structural change should also participate [15,46]. Importantly, the recent crystallographic data indicated that the difference between these two ferryl forms is in the slight variance of the axial coordination of Fe^4+^ [47].

Interestingly, another **F**-type ferryl form (**F^•^**) is possible to prepare by reacting the **O** with one molecule of H_2_O_2_ in acidic buffers [37,39,43]. Based on the nature of the reaction and the apparent proton uptake into the Fe_a3_-Cu_B_ center during this process [39,43], the production of the **F^•^** may be illustrated by this scheme:[Fe_a3_^3+^-OH HO-Cu_B_^2+^ YOH] + H_2_O_2_ + H^+^ → [Fe_a3_^4+^=O H_2_O-Cu_B_^2+^ YO**^•^**] + 2H_2_O(3)
**O              F**

Small differences between the **F^•^** (a dot underline the radical present at the catalytic site of CcO) and the **F** intermediates in UV-Vis absorption spectra may indicate the distinct redox state of the catalytic center in these two ferryl states [37,39,43,48,49,50]. Particularly, the position of the maxima of the α-band at 575 and 580 nm are observed in the difference spectra of the **F^•^** and the **F** vs. the **O**, respectively.

Several previous studies, employing the electron paramagnetic resonance (EPR), uncovered multiple radicals in the peroxide-generated ferryl forms of CcO, however, showing low and variable yields [37,51,52,53,54]. Moreover, the correlation between the amount of the detected radicals and the concentration of the **F^•^** form of the bovine CcO has not been established, yet [51,53,54]. In addition, some earlier works indicated the migration of the primary radical a large distance from the heme *a*_3_-Cu_B_ center [55,56,57,58,59]. When the oxidized enzyme was reacted with an excess of H_2_O_2_, the oxidative modification of the distant tryptophan residues and the bound phospholipids together with the spin trapping of the radical at the surface of purified CcO has been observed [55,56,57,58,59].

In this study, the formation of the radical in the **F^•^** state of the purified bovine CcO was verified and its lifetime at the catalytic center was examined. The data obtained by the application of four different experimental approaches support the production of the radical at the catalytic center in the **F^•^** form. However, autoxidation of CcO triggers the quenching of this primary radical with a rate that should be at least ten times faster (time constant of τ ≤ 9 s) than the rate of the spectral development (τ ≈ 90 s) of the ferryl state of heme *a*_3_ (Fe_a3_^4+^=O) (pH 5.7, 5 °C). The data also showed that the small spectral differences between the **F^•^** and the **F** states do not indicate the distinct redox state of the catalytic center in these ferryl forms.

## 2. Results

In all measurements, the **F^•^** intermediate was produced by the reaction of the oxidized CcO (**O**) with either sub- or stoichiometric amount of H_2_O_2_ in an acidic buffer at pH 5.7 and 5 °C. The formation of **F^•^** and its endogenous conversion to the **O** was monitored by a combination of isothermal titration calorimetry (ITC), UV-Vis absorption, and EPR spectroscopies.

### 2.1. Formation and Decay of F^•^ Form—UV-Vis and ITC Measurements

The production of the **F^•^** state by the reaction of the oxidized CcO (49 μM) with a sub-stoichiometric amount of H_2_O_2_ (5.4 μM) (**O** + H_2_O_2_ → **F^•^**) is shown in Figure 3. The kinetics of the formation of this state followed by its spontaneous transition to the **O** form were registered using the change of the absorbance ΔA (575–630 nm) (Figure 3A). After the addition of peroxide, the production of the **F^•^**, represented by the rise of ΔA (575–630 nm), is completed in ~600 s. The subsequent decline of the absorbance is due to the endogenous decay of the **F^•^**-to-**O** state. From the two-exponential fit of this kinetics, the time constants of ~80 s and ~3 × 10^3^ s were attained for the generation and the decomposition of the **F^•^**, respectively.

The production of the **F^•^** state is shown by the difference spectrum, **F^•^**
*versus*
**O (F^•^-O**), exhibiting the characteristic maxima at 575, 534, and ~435 nm and the minimum at ~412 nm (Figure 3B). The spectrum was collected at the time when the maximal amount of the ferryl form was formed (600 s). The average amount of the **F^•^** produced by a single molecule of H_2_O_2_ was found to be 1.02 ± 0.07 (n = 3).

A large noise in the difference spectrum in the region of the Soret band (400–450 nm) is a consequence of the high absorbance in this region (ca. 1.5). In the absolute spectrum, this band exhibits the maximum at ~423 nm for the oxidized CcO and is slightly red-shifted (~424 nm) after the reaction with peroxide.

The heat changes associated with this reaction are illustrated in Figure 4 (**F^•^**, full line). The negative values of the time dependence of the power demonstrate that heat is released in this reaction. The maximum rate of heat release is observed at the beginning of the reaction initiated by the peroxide injection (Figure 4, H_2_O_2_ arrow). At about 600 s, when the formation of the **F^•^** is spectrally completed, this rate is nearly zero. From the integration of the area under the ITC curves, the average reaction enthalpy change ΔH = −39.2 ± 0.7 kcal/mol H_2_O_2_ (n = 3) was obtained.

Control ITC measurements during the reaction of 50 μM CN-ligated CcO with 5 μM H_2_O_2_ demonstrate that there is a negligible contribution of possible side reactions (Figure 4, CcO.CN line). In this case, the heat released is about 2% of the value of ΔH obtained for the uninhibited CcO. This heat was subtracted from the measured value of ΔH.

A similar ferryl state of heme, the compound I (Fe^4+^=O π**^•+^)**, can be generated by reacting the oxidized horseradish peroxidase (HRP) with the stoichiometric amount of H_2_O_2_ (Figure 4, cmpI dashed line). The advantage of this measurement is that the presence and stability of both the ferryl iron and the π cation radical can be simply monitored by the optical spectrum [60,61]. The obtained spectra showed that the reaction of 5 μM of the oxidized HRP with 5 μM H_2_O_2_ produces in mixing time (~10 s) a stable compound I at pH 5.7 and 5 °C. The enthalpy change ΔH of −27.1 ± 1.8 kcal/mol H_2_O_2_ (n = 3) was associated with this reaction. In the determination of the reaction ΔH, the heats of dilution and mixing were subtracted.

### 2.2. Formation of Radical in the F^•^ State—UV-Vis and EPR Measurements

The production of the **F^•^** state and its endogenous transition to the **O** was registered by the parallel measurements of its visible absorption and EPR spectra. The time dependence of the formation and decay of **F^•^**, **P_M_,** and radical (**R^•^**) observed by EPR spectroscopy in the reaction of 98 μM **O** with 98 μM H_2_O_2_ is presented in Figure 5. The time dependencies of the relative amount of **F^•^**, **P_M_,** and **R^•^**
*versus* the total concentration of the **O**, are shown for the time period of 6 × 10^3^ s. The maximum amount of the **F^•^** is observed in ~1000 s after the addition of peroxide to the **O** state (Figure 5A, **F**). The appearance of the **F^•^** form is, however, attained in two different kinetic phases. The first and major phase is completed in ~120 s (70%) and is characterized by a time constant of ~26 s. The next slower phase is described by τ = ~330 s (30%) (Table 1). Subsequently, the much slower endogenous transition of the **F^•^**-to-**O** state proceeds with τ = 2.9 × 10^4^ s.

At the same time, the transient appearance of a small fraction of the **P_M_** state is also detected in this sample (Figure 5A, **P_M_** dotted line). The time constant of this phase, ~22 s, is close to the rate of the rapid phase of the formation of the **F^•^**. The disappearance of the **P_M_** occurs with the same time constant of ~330 s as the rate of the generation of the **F^•^** in the slow phase (Table 1).

The difference in visible spectra also demonstrates the initial rapid appearance of the **P_M_** state (Figure 5B). This form is distinguished by the band having the maximum at 607 nm in the spectrum registered 120 s after the initiation of the reaction (Figure 5B, 120 s full line). This band at 607 mm then disappears and only a single band with the maximum at 575 nm is present (1000 s after injection of peroxide, dashed line).

The radical in the EPR spectra appear with the time constant of τ = ~200 s followed by its much slower disappearance (τ = ~3300 s) (Figure 5A, **R^•^** dashed line). However, its maximal yield, calculated relative to the EPR signal of Cu_A_^2+^, is only about 0.2% of the used CcO (0.6% relative to the formed **F^•^**) (Figure 5C). This is documented by the EPR spectra of the initial **O** and the **F^•^** forms collected on the sample frozen in 1200 s after the peroxide addition. The spectrum of the radical (Figure 5C, **F**-**O**) was obtained by the subtraction of the spectrum of the **O** from the **F^•^** state. The radical is characterized by g = 2.005 and the linewidth of ~28 G.

### 2.3. One-Electron Reduction of the F^•^ Form

The redox state of the catalytic center of CcO in the **F^•^** state was examined by the reduction with the substoichiometric amount of one-electron donor to minimize more than one electron transfer into the catalytic center of CcO. The formation of the **F^•^** state and its reduction by ferrocyanide under aerobic conditions are shown in Figure 6. At the time of the addition of catalase (Figure 6A, cat arrow), 300 s after the peroxide injection, the interaction of the **O** (50 μM) with the substoichiometric concentration of H_2_O_2_ (48 μM) resulted in the formation of 19.3 μM **F^•^** (39%) with the kinetics characterized by τ = ~90 s.

Into this sample, containing 19.3 μM **F^•^** and 30.7 μM **O**, the sub-stoichiometric amount of potassium ferrocyanide (9.9 μM, final concentration) was injected as indicated by the second arrow (Figure 6A, Fe^2+^ arrow). The injection of the electron donor triggers a rapid disappearance of the **F^•^** displayed as the fast decrease of ΔA (575–630 nm) (Figure 6A) followed by a slow endogenous transition of the **F^•^**-to-**O**.

The difference between the spectrum collected before and 500 s after the addition of ferrocyanide (Fe^2+^) revealed that in this sample some small fraction of the **P_M_** is also present. The reduction of these two ferryl forms is demonstrated by the characteristic maxima at 575 nm for **F^•^** and 607 nm for the **P_M_** state (Figure 6B). In this case, the loss of 5.1 μM of **F^•^** and 2.7 μM **P_M_** was detected. The average molar amount of lost both ferryl forms ([**P**] + [**F^•^**]) relative to the utilized ferrocyanide was 0.83 ± 0.1 (n =3).

## 3. Discussion

The pH-dependent production of two types of ferryl forms in the reaction of the oxidized bovine CcO with a single molecule of H_2_O_2_ has been explained by the following branching Figure 1 [39,43]:

In this scheme, the reversible binding of peroxide into the catalytic center (**P_0_** intermediate) is followed by the irreversible redox reaction. The **P_0_** intermediate is very likely single or double-ionized hydrogen peroxide bound to heme a_3_ forming a non-stable iron-peroxy adduct, Fe_a3_^3+^-O-OH (or Fe_a3_^3+^-O-O) analogous to compound 0 of peroxidases. At basic pH values dominates the **P_0_**-to-**P_M_** conversion (**P_0_** → [Fe_a3_^4+^=O HO-Cu_B_^2+^ YO^•^]). The transition of the **P_0_**-to-**F^•^**, occurring in acidic solutions, is very likely due to the simultaneous uptake of one proton into the catalytic center (**P_0_** + H^+^ → [Fe_a3_^4+^=O H_2_O-Cu_B_^2+^ YO^•^]) [39,43]. It means that both the **P_M_** and the **F^•^** forms should be two oxidizing equivalents above the oxidized CcO. In the absence of external electron donors, these two ferryl states relax very slowly to the apparent oxidized CcO by the autoxidation of the enzyme [55,57,62,63].

The obtained data are in the agreement with the formation of a radical in the peroxide-produced **F^•^** state. More importantly, the results indicate that the lifetime of the primary YO^•^ radical in the **F^•^** is much shorter than that of the ferryl Fe_a3_^4+^=O iron. At the time when the ferryl heme *a*_3_ state is developed, the radical in the catalytic center is almost fully quenched. This conclusion is substantiated by three different experimental observations: the enthalpy changes, the EPR detection of the radical, and the reduction of the **F^•^** form.

The first indication is the large enthalpy change (−39 kcal/mol) associated with the **O**-to-**F^•^** transition contrasting with that of the generation of compound I of HRP (−27 kcal/mol) (Figure 4). In the reaction of the oxidized CcO and HRP with H_2_O_2_, the measured ΔH values can be attributed to two identical events: one is the reversible H_2_O_2_ binding to the oxidized heme iron and the other is the redox reaction. The earlier investigations of the peroxide binding to the different oxidized heme proteins, employing the mutant of human myoglobin (His64Gly) [64], Mn-reconstituted myoglobin [65], horseradish peroxidase at subzero temperatures [66], and Mn-reconstituted horseradish peroxidase [67], revealed only a small positive ΔH with the values between zero and +4 kcal/mol H_2_O_2_.

Accordingly, the measured large negative ΔH values can be attributed mostly to the redox reactions (**P_0_** → [Fe^4+^=O R**^•^**]). However, from these two heme proteins we know only for the compound I of HRP that both the ferryl iron and π-cation radical are present and very much stable during the ITC measurements. Then the measured ΔH of −27 kcal/mol represents the generation of the [Fe^4+^=O R^•^] state without the contribution of possible side reactions. Because of the similarity of the reactions and the reduction potentials of the ferryl iron and radical in both proteins [10,60,68,69,70] it may be also expected a similar value of ΔH for the production of the **F^•^** state. Consequently, the excess heat released during the production of the apparent **F^•^** state (−39 kcal/mol) implicates some additional reaction (s).

Multiple experimental observations indicate that the side reaction (s) responsible for the excess heat liberated during the generation of the **F^•^** state is very likely the radical migration and its quenching. This explanation is supported by the observations of the different types and amounts of radicals [37,51,52,53,54] when the oxidized CcO was reacted with excess H_2_O_2_. More importantly, it has been shown by the EPR spin-trapping technique that H_2_O_2_ interaction with the catalytic center of the oxidized CcO leads to the appearance of the cysteine thiyl radical [58] and lipid-based radical [63]. Similarly, the spin-trapping using CcO/organic hydroperoxides system documented the protein-centered radical and possible participation of the surface-positioned Tyr, Trp, and Cys residues in its migration [59]. The large distance translocation of the radical, 30–60 Å from the heme *a*_3_-Cu_B_ center, is corroborated by the modification of several distant Trp residues and bound phospholipids in the H_2_O_2_/CcO system [55,56,57]. The radical migration through the protein matrix is very likely facilitated by ‘wires’ composed of aromatic amino acid residues [56,71].

Secondly, the parallel visible absorption and EPR measurements uncovered only a very small amount of the free radical (~0.6%) relative to the formed **F^•^** state (Figure 5). This amount is even smaller than previously reported (5–20%) for bovine CcO [37,51,53,54]. This difference can be understood since the nature and the amount of the radical is dependent on the concentration of H_2_O_2_, pH of solutions, and very likely also on the reaction time [37,39,50,51,53]. Typically, two overlapping radical signals are observed, one described with a line width of 45 G and another with a width of 12 G. These two signals have been recently assigned to two Tyr radicals. Narrow one to Tyr244, covalently bound to His240 (ligand of Cu_B_), and the second to Tyr129 located in the proximity of this center (~10 Å) [53,54]. However, the observed radical in this work showing g = 2.005 and the line width of ~28 G cannot be attributed to the combination of the 45 and 12 G-signals. It seems that this radical is composed of two derivative-like signals: one belongs to the previously detected 12 G and the other is characterized by the width of ~30 G. Identity of the broader signal is presently not certain.

The kinetics of the appearance of the radical (τ = ~200 s) does not correlate with the generation of the **F^•^** state (Figure 5, Table 1). The appearance of the radical is, however, closer to the kinetics of the conversion of the small fraction of **P_M_**-to-**F^•^** state (slow phase, τ = ~330 s) (Figure 5, Table 1). Previously, the kinetics of the conversion of the **P_M_**-to-**F^•^** was found to correlate with the appearance of a small amount of the radical [51]. In this earlier work, the **P_M_** state was produced by the reaction of two-electron-reduced CcO with O_2_ in a basic buffer, and the **P_M_**-to-**F^•^** conversion was initiated by the rapid acidification of the solution. The observed radical was attributed to the species formed by the migration of the primary radical from the catalytic center of CcO.

It is believed that the missing detection of the radical in either the **F^•^** or the **P_M_** states should be a consequence of the nearby metal ions (Fe_a3_^4+^=O, Cu_B_^2+^). The spin coupling of the radical with Cu_B_^2+^ or the broadening of the radical signal by the interaction with these paramagnetic centers could be a reason for its absence in the EPR spectrum [37,51,72]. Another possibility, indicated by several earlier studies [55,56,57,58,59] together with the present work, is that the primary radical is unstable and can migrate to the surface of the protein and be quenched or captured by spin traps.

Our third approach, a verification of the redox state of the **F^•^** form, corroborates the rapid migration of the primary radical and its quenching (Figure 6). The reduction of the **F^•^** state by the sub-stoichiometric amount of ferrocyanide showed that the molar ratio of the lost **F^•^** along with the **P_M_** state ([**F^•^**] + [**P_M_**]) to the used ferrocyanide is close to one (0.83 ± 0.1). If both the **F^•^** and the **P_M_** were two oxidizing equivalents above the **O** then this ratio should be 0.5. The result demonstrates that at the time of examination, one-electron reduction of any of these two ferryl intermediates leads to the formation of the oxidized CcO. This time-dependent process is summarized in this scheme: [Fe_a3_^3+^-OH HO-Cu_B_^2+^ YOH] + H_2_O_2_ → [Fe_a3_^4+^=O H_2_O-Cu_B_^2+^ YO^•^] → [Fe_a3_^4+^=O H_2_O-Cu_B_^2+^ YOH](4)
** O              F^•^** (575)              **F** (575)
where the reaction of **O** with one molecule of H_2_O_2_ generates the **F^•^** state having the ferryl iron of heme *a*_3_ together with the radical, probably Tyr244, at the catalytic center. In the absence of external electron donors, the formation of **F^•^** is followed by the autoxidation of the enzyme that results in more rapid quenching of the radical (YO•). However, the quenching of radical and production of the **F** state are not associated with observable changes in the optical spectrum. Thus, the prepared sample is a mixture of **F^•^** and **F** states showing the maximum at 575 nm and whose composition is dependent on the reaction time.

The data showed that the maxima at 575 nm and 580 nm in the difference spectra of the **F^•^** and **F**, respectively, [37,39] are not caused by the presence or absence of radicals at the catalytic center. We have noticed that the maximum at 575 nm results from the pH dependence of the spectrum of the oxidized CcO [73] utilized in the calculation of the difference spectrum. If the spectrum of the oxidized CcO at pH 8.0 is substituted into this calculation (**F^•^**-**O**), the maximum at ~580 nm is observed.

The rate of the radical quenching at the catalytic site in CcO can be estimated from its absence at the time when the **F^•^** form is developed (Figure 6). Since the formation of the Fe_a3_^4+^=O state in the **F^•^** form takes place with the time constant of ~90 s, then the radical quenching should be at least ten times faster. Consequently, the radical will be lost if its quenching will occur with a time constant of at least less than 9 s (pH 5.7, 5 °C).

The behavior of the primary radical in the **F^•^**, a shorter lifetime relative to the ferryl iron, is very the same as that what we have recently observed for the radical in the **P_M_** form at pH 8.0 [74,75]. We have found that the **P_M_** form is converted with elapsed time to the spectrally similar **P_R_** form with no radical at the catalytic center [74,75].

The instability of the primary radical in the ferryl **F^•^** and **P_M_** forms of CcO is, however, not exceptional among the heme proteins. Such behavior of the radical is very analogous to that of the primary radicals in myoglobins [76,77,78,79,80], hemoglobins [79,80,81,82], cytochrome *c* peroxidase [83], ascorbate peroxidase [84], and prostaglandin H synthase [85].

Ultimate intrinsic electron donors of electrons for the reduction of radicals appear to be Trp residues and phospholipids bound to the purified CcO [55,56,63]. The oxidation of Trp residues and lipids to conjugated dienes and trienes initiated by the reaction of H_2_O_2_ with the catalytic center of oxidized CcO has been demonstrated. Cys [58] and Met residues, with midpoint potentials around −250 mV, may also serve as a source of electrons.

The radical migration to the surface of the ferryl form also brings a new explanation for the release of superoxide when oxidized CcO is reacted with an excess of H_2_O_2_ [38,45]. Since the production of superoxide was inhibited by the binding of cyanide to the Fe_a3_-Cu_B_ center it was assumed that O_2_**^•−^** is released by the direct reduction of the radical at the catalytic center by peroxide. However, the production of O_2_**^•−^** can be also explained by the reaction of H_2_O_2_ with the radical migrated to the surface of the ferryl intermediate. In this case, the binding of cyanide to the catalytic center of CcO will also prevent the production of superoxide. For additional confirmation of the rapid radical migration from the catalytic center to the surface of the enzyme, the EPR spin-trapping technique will be employed.

## 4. Materials and Methods

### 4.1. Materials

Potassium phosphate monobasic and dibasic, potassium hydroxide, potassium ferricyanide, potassium ferrocyanide, potassium sulfate, potassium cyanide, horse heart cytochrome *c*, superoxide dismutase (SOD), horseradish peroxidase type VI A, and catalase from bovine liver were purchased from Sigma-Aldrich, Triton X-100 (TX) was from Roche Diagnostics, dodecyl maltoside (DM) from Anatrace, Sepharose Q fast flow from Pharmacia Uppsala and hydrogen peroxide solution (~30%) was from Fluka.

Bovine heart cytochrome *c* oxidase was isolated from mitochondria following the modified method [86] into 10 mM Tris, pH 7.6, 50 mM K_2_SO_4_, and 0.1% TX. To change TX for DM detergent, the purified enzyme was diluted and reconcentrated using microfilters (YM 100, Millipore, cut-off 100 kDa) with the buffer containing 0.1% DM.

Isolated CcO was frozen in liquid nitrogen and stored at −80 °C. The concentration of CcO was determined from the UV-Vis absorption spectrum of the oxidized enzyme using an extinction coefficient ε (424 nm) = 156 mM^−1^cm^−1^ and ε (428 nm) = 169 mM^−1^cm^−1^ for the cyanide-ligated CcO (CcO.CN) [87].

### 4.2. Preparation of Oxidized CcO

The purified CcO may contain some small fractions of a partially reduced enzyme. To obtain the fully oxidized CcO the isolated enzyme (150–200 μM) was incubated with 10 mM ferricyanide for 10 min at room temperature. The samples were then passed through a desalting PD-10 column utilizing 5 mM potassium phosphate buffer (KPi), 50 mM K_2_SO_4_, pH 8.0 containing 0.1% DM.

The CcO.CN complex, without free cyanide in the solution, was produced in 20 min. incubations of the purified CcO with 10 mM KCN at 4 °C followed by the addition of 10 mM ferricyanide. This sample was desalted on the PD 10 column 600 s after the addition of ferricyanide as it is described above.

### 4.3. Isothermal Titration Calorimetry (ITC) Measurements

The enthalpy changes (ΔH) associated with the formation of the **F^•^** state of CcO and compound I of horseradish peroxidase (HRP) were measured by the ITC method during the reaction of the oxidized proteins with sub- and stoichiometric concentrations of H_2_O_2_. The ITC cell was filled with the oxidized protein (~5 μM HRP and ~50 μM CcO) and the reaction was initiated by a single injection of peroxide (5 μM, final concentration). The same buffer was used for the protein in the cell and H_2_O_2_ in the injection syringe. Typically, 1.3 μL of H_2_O_2_ was injected into the cell in 2 s. The measurements were performed in a MicroCal ITC 200 (GE) instrument at the temperature of 5 °C.

The heat linked with the non-specific reactions that could take place in the course of the formation of the **F^•^** state of CcO was assessed by the injection of H_2_O_2_ into the cell filled with the oxidized CcO ligated with cyanide. The bound cyanide at the catalytic Fe_a3_-Cu_B_ center of CcO blocks the production of the ferryl state. These control measurements were carried out under identical conditions as those used for the determination of the ΔH during the reaction of the oxidized protein with H_2_O_2_. Dilution and mixing heat (~3–6% of the total ΔH) were subtracted from the measured reaction enthalpies.

In all experiments in this study, potassium phosphate buffer was used because of its low ionization enthalpies which were important for the measurements of reaction enthalpies by ITC. The temperature of 5 °C was selected in these measurements to decelerate the formation of the **F^•^** state and to increase the stability of the radical in this form.

### 4.4. UV-Vis Absorption Spectroscopy Mesurements

The formation of the **F^•^** state, its spontaneous decay, and as well the reduction of this form by ferrocyanide was monitored by visible absorption spectroscopy. The spectral changes induced by the addition of peroxide to oxidized CcO were registered in the diode array spectrometer (Specord S600) at 5 °C. The spectra were recorded in the range of 400–700 nm every 10 s for the first 600 s and every 20 s for the next 3 × 10^3^–6 × 10^3^ s. From the accumulated spectra the kinetics of the absorbance changes at certain wavelengths and the spectra at the given reaction time was attained.

The reaction of the oxidized CcO with one molecule of H_2_O_2_ at acidic buffers results in the production mostly of the **F^•^** form as well as a transient and minor population of the **P_M_** state. The concentrations of these two ferryl intermediates were calculated from the difference spectrum of the peroxide-treated CcO minus the oxidized CcO employing Δε (607–630 nm) = 11 mM^−1^cm^−1^ for the **P_M_** [88]. Since the **P_M_** makes also a contribution to the absorbance change ΔA (575–630), the measured value was corrected using Δε (575–630 nm) = 2.1 mM^−1^cm^−1^. After this subtraction, the concentration of the **F^•^** was calculated using Δε (575–630 nm) = 5.3 mM^−1^cm^−1^ [88].

The transient rise and the subsequent decay, representing the formation and the decomposition of the ferryl states and the radical, were fitted by the two-step A → B → C model:k_1_ k_2_
A → B → C
using equation B = A_0_(((k_1_/(k_2_ − k_1_))(e^−k1t^/e^−k2t^). Where k_1_ and k_2_ are corresponding rate constants and A_0_ is the initial concentration of the enzyme participating in the reaction.

Extinction coefficient ε (240 nm) = 0.04 mM^−1^cm^−1^ was utilized to determine the concentration of H_2_O_2_ [89] and ε (420 nm) = 1 mM^−1^cm^−1^ of ferricyanide [90]. The concentration of ferrocyanide was established by the oxidation of ferrocyanide to ferricyanide by KMnO_4_. From the spectrum of the produced ferricyanide, the concentration of ferrocyanide was calculated.

### 4.5. EPR Spectroscopy

The EPR spectra were collected on the frozen samples of 98 μM oxidized CcO reacted with 98 μM H_2_O_2_ at 5 °C in 100 mM phosphate buffer, pH 5.7, containing 30 mM K_2_SO_4_, 0.6% DM, and 267 units/mL of SOD. After the addition of peroxide to CcO, the aliquots (~0.3 mL) were taken at a certain time from the incubated stock solution and frozen rapidly in a methanol/dry ice bath, and then transferred to liquid nitrogen. These samples were stored in liquid nitrogen until the measurements were performed.

The measurements were carried out in Bruker Elex Sys E500 spectrometer at 80 K using microwave power of 0.2 mW. Other parameters of the measurements were: microwave frequency 9.39 GHz, modulation amplitude 10 G, modulation frequency 100 kHz, and time constant 164 × 10^−3^ s.

The generation of the **F^•^** state from the oxidized CcO (Figure 3), the ITC measurements (Figure 4), and the reduction of the **F^•^** form by ferrocyanide (Figure 6) represent the typical outcome of three measurements in each case. The EPR examination of the radical in the **F^•^** form shown in Figure 5 is representative of two measurements.

## 5. Conclusions

In this work, the redox state of the catalytic heme *a*_3_-Cu_B_ center of the peroxide-produced **F^•^** ferryl form of bovine CcO and the stability of the formed radical in this form were investigated for the first time. The comparison of the heats liberated during the formation of the **F^•^** in the reaction of the oxidized CcO with H_2_O_2_ and that of the compound I of horseradish peroxidase indicates the creation of a radical at heme *a*_3_-Cu_B_ center. However, the primary radical in the **F^•^** state, plausibly Tyr244, exhibits a much shorter lifetime (τ ≤ 9 s) relative to the rate of the production of the accompanying ferryl iron of heme *a*_3_ (τ = ~90 s). The EPR data, the large enthalpy changes, and the reduction of the **F^•^** form by the defined amount of ferrocyanide showed that the migration of the radical from the catalytic center is associated with its rapid quenching, very likely at the surface of the protein.

## Data Availability

Date are presented in this manuscript.

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
