# Peer review of "Radical in the Peroxide-Produced F-Type Ferryl Form of Bovine Cytochrome c Oxidase"

_ijms, 2022, doi:10.3390/ijms232012580_

Round 1
Reviewer 1 Report
In this paper Fabian, Jancura and co-workers have investigated the formation of a F-type ferryl form derived from the reaction of the oxidized CcO with sub- or stoichiometric amount of H2O2 in acidic condition. In these experimental conditions the intermediate producted, F•, contains a free radical at the heme a3-CuB center.
The authors examined the formation and the endogenous decay of both the ferryl iron of heme a3 and the radical in F• intermediate by different experimental techniques such as isothermal titration calorimetry, electron paramagnetic resonance, and electronic absorption spectroscopy.
This is an interesting study and a lot of work has been carried. Even though the idea behind this manuscript is not completely new, it contributes to elucidate the intermediate structures in the CcO catalytic cycle, and to better understand both the mechanism of oxygen reduction and its coupling to proton pumping.
However, the manuscript is, in my opinion, difficult to read and it may be very hard for the reader to follow the discussion. Furthermore, the set of the names and abbreviations currently used to denote the catalytic intermediates of CcO is not explained and it may be confusing for a reader who is non-specialist in this topic.
Thus, at the moment, I recommend a significant re-writing, before publication.
Major points
- Figure 2 (pag. 2) shows the catalytic cycle of cytochrome c oxidase and the authors discuss the states of the catalytic cycle. In the literature are reported catalytic cycles with different intermediates. Please, clarify this point.
2. On page 3 the authors state: “ Interestingly, another F-type ferryl form (F•) is possible to prepare by reacting the O…….”
The F• and PM intermediates show different spectroscopic features (PM shows a sharp a-band with a maximum at 607 nm, while the maximum of the a-band for F• is at 575 nm), but from scheme (3) and (4) the chemical formulas for F• and PM are very similar. It is true? Do the authors have any information about that?
3. The intermediate F• was produced in acidic buffer (phosphate buffer at pH = 5.7) and at T = 5°C.
From the text it is not clear how were the buffer and the temperature selected. Did the authors evaluate the effect of buffer and of temperature on the intermediate formation?
4. Figure 3A shows the kinetics of the formation of F• form. Why the absorbance change ΔA(575–630 nm) was used to monitor the formation of this ferryl state. Is 630 nm a reference point? Did the authors acquire the UV-vis spectrum of the F• form in the spectral range 400-700 nm? If yes, what is the position of the maximum of the Soret band? It is not clear to me why the difference spectrum F• versus O (Figure 3B) is not very resolved in the Soret region.
5. On page 6, the authors state: “ … The generation of F•, PM and radical (R•) in the reaction ……”. What is radical (R•)? Please, specify.
6. On page 7, Table 1, the authors report the time constants of the generation of ferryl intermediates. How were derived the time constants? What is the equation used?
7. The scheme on pag. 9 shows a P0 form and the authors discuss (lines 227-234) on the transitions, between the different intermediates, that occur at different pH values. What are the features of the P0 compound?
Minor points
Several experimental procedures are not clearly described in the material and methods section. In particular:
- On page 12 line 396-397, the authors state: “The conditions in these spectral measurements were identical to those used in the ITC data collections.” However the experimental data are not reported in the ITC measurements paragraph.
- Page 6, line 159 (Fig. 5A, F) and not (Fig. 5, F).
- Page 6, line 174 (Fig. 5A, PM dotted line) and not (Fig. 5, PM dashed line).
- Page 7, line 188 (Fig. 5A, R• dashed line) and not (Fig. 5, R• dashed line).
Author Response
We would like to thank the reviewers for the concerns, questions and suggestions and that helped us to improve the presentation of our data as well as to clarify some aspects of the manuscript. The manuscript has been revised with respect to the reviewer’s comments and a list of all changes is provided in the following paragraphs.
Reviewer 1
However, the manuscript is, in my opinion, difficult to read and it may be very hard for the reader to follow the discussion.
We tried to identify a possibly complicated or difficult parts of the Discussion and made corresponding corrections to improve readability.
However, it was quite surprising to learn that whole discussion is hard to follow. Working on the manuscript we took a great care to make a structure and argumentation in Discussion as simple as possible. In addition one figure and several reaction schemes summarized our data and interpretation that offer a large assistance in the understanding.
In spite of this opinion we still believe that the discussion is quite straightforward. Reasons are following.
- 1. At the very beginning there is a scheme showing only two possible ferryl states, PM and F , produced by the reaction of CcO with single molecule of H2O2 and having radical at the catalytic center. The presence of this form is pH dependent at the acidic pH values the F• state is dominating.
- 2. To make a text more comprehensible the main conclusion of this study is presented in the opening of discussion. It is stated that the formation of the ferryl Fintermediate is associated with the rapid loss of the radical from the catalytic center very likely by the migration to the surface of the enzyme followed by its quenching.
- 3. For support of this conclusion are presented and discussed three observations, the enthalpy changes, the EPR detection of the radical and reduction by ferrocyanide. The following discussion is in the order of the presentation the results.
3A. It is examined a large release of the heat during the formation of the F• state. Based on our and published data it is attributed to the instability of the radical in this form.
3B. This suggestion is next discussed in the relation to the EPR data. It is described that a very small yield of free radical observed in the course of formation and the endogenous decay of the F• is in the favor of this interpretation. Because of the slight yield of the radical and the missing kinetic correlation with appearance of the ferryl state of heme a3 it is proposed that the migrated radical form the catalytic center is also quenched. Usually the absence of the quantitative EPR signal of radical has been attributed to the spin coupling or the rapid relaxation because of the close proximity of the paramagnetic centers.
3C. Finally, the observation that one electron is able to convert the F• to oxidized CcO is discussed. It is consistent with the previous two facts, the heat release and EPR detection of the radical, and shows that the radical has to be almost absent in the F• at the time when this from is spectrally developed.
This is the main frame of the Discussion and we believe that it is written in a comprehensible way for a reader and provides summarization and interpretation of the data.
Furthermore, the set of the names and abbreviations currently used to denote the catalytic intermediates of CcO is not explained and it may be confusing for a reader who is non-specialist in this topic.
The explanation of the names and abbreviations used to denote the ferryl intermediates of the catalytic cycle in CcO is now given in the Introduction (lines 44-63). Concretely,
Very likely from four ferryl intermediates only states denoted as PM and the PR might bring some confusion and make more difficult to follow the discussion. However, these names are still used in literature for a historical reason.
The PM is produced by the reaction of two-electron reduced (Mixed valence) CcO with O2 [11-15]. It was named PM since originally it was expected to have a peroxide bound at the catalytic center (e.g. Fea33+-O-O-CuB2+). However, it has been shown later that the dioxygen bond in this intermediate is already cleaved by a four-electron reduction of O2. Two of them are coming from the oxidation of Fea32+ to Fea34+=O, one from the oxidation of CuB+ to CuB2+ and the donor of the fourth electron is plausibly the nearby Tyr244 (bovine CcO numbering), producing the free radical (YO•) at the vicinity of the catalytic center (Figs. 1 & 2).
The name for the PR state originates from observation that delivery of the third electron into the catalytic site of PM leads to the reduction of YO• radical with almost no changes in the visible spectrum. Since the radical is reduced in this intermediate and its visible absorption spectrum is similar to PM it was named PR [12, 16-19].
Very similar relation is between the F and F• intermediates. The radical is present in F• (dot underlines the radical at the catalytic center) and absent in the F.
This explanation is now given in the Introduction (lines 92-94).
Major points
- Figure 2 (pag. 2) shows the catalytic cycle of cytochrome c oxidase and the authors discuss the states of the catalytic cycle. In the literature are reported catalytic cycles with different intermediates. Please, clarify this point.
Intention of Figure 2 (page 2) was not to show the complete catalytic cycle of cytochrome c oxidase. This figure illustrates only the sequence of the ferryl intermediates appearing in the catalytic cycle which are related to this study. For the sake of simplicity, several catalytic intermediates were omitted in this scheme. To avoid an impression that the figure shows all intermediates of the catalytic cycle of CcO, the figure legend was corrected (page 2).
- On page 3 the authors state: “Interestingly, another F-type ferryl form (F•) is possible to prepare by reacting the O…….”
The F• and PM intermediates show different spectroscopic features (PM shows a sharp a-band with a maximum at 607 nm, while the maximum of the a-band for F• is at 575 nm), but from scheme (3) and (4) the chemical formulas for F• and PM are very similar. It is true? Do the authors have any information about that?
Certainly, the UV-Vis absorption spectroscopic differences between P and F states cannot be assigned simply to proton uptake. Recent crystallographic data indicated that the P to F transition is accompanied with the change of the angle between the bonds of the nitrogen atom of His376 coordinated to Fea34+ and the bound oxygen atom (N – Fea34+=O) (79). The bent structure, characterized by the angle of 175°, in the P state is changed to the straight coordination with the angle of 180° in the F state. The notion about this result is now presented in the Introduction section (lines 81-85).
- The intermediate F• was produced in acidic buffer (phosphate buffer at pH = 5.7) and at T = 5°C.
From the text it is not clear how were the buffer and the temperature selected. Did the authors evaluate the effect of buffer and of temperature on the intermediate formation?
Answer: In all performed experiments in this study, potassium phosphate buffer was used because of its low ionization enthalpies that was important for the measurements of reaction enthalpies by ITC. Reaction temperature 5 °C was selected to decelerate the formation of F• state and to increase the stability of the radical in this intermediate.
Reason to use phosphate buffer at 5 °C is now presented in the Material and Methods (lines 429-432).
- Figure 3A shows the kinetics of the formation of F• form. Why the absorbance change ΔA(575–630 nm) was used to monitor the formation of this ferryl state. Is 630 nm a reference point? Did the authors acquire the UV-vis spectrum of the F• form in the spectral range 400-700 nm? If yes, what is the position of the maximum of the Soret band? It is not clear to me why the difference spectrum F•versus O(Figure 3B) is not very resolved in the Soret region.
- To monitor F• state the absorbance changes DA(575-630 nm) were utilized since for this couple of the wavelengths the extinction coefficient for F type ferryl form is very well established and at 630 nm only a slight change of the absorbance occurs during the transition of the oxidized CcO to the F• form.
- Spectra were registered between 400 and 700 nm as it is described in the Materials and Methods (lines 438-439). The maximum of the Soret band of the oxidized CcO and F• form was at ~423 nm and ~424 nm, respectively. These data are now provided in the revised manuscript.
- The large noise in the difference spectrum in the region of the Soret band is a consequence of a high absorbance in this range (above 1.5). We intentionally sacrificed the high quality of the Soret band spectrum to get the reliable spectra in the range between 500 and 700 nm. This wavelength range is most important for the discrimination and quantification of the ferryl intermediates. This is explained in the revised manuscript (lines 145-148).
- On page 6, the authors state: “ … The generation of F•,PMand radical (R•) in the reaction ……”. What is radical (R•)? Please, specify.
Symbol R• simply represents the radical detected in the EPR spectra during the reaction of the oxidized CcO with H2O2 (Fig. 5). The symbol R• is now specified in the Results section (line 185).
- On page 7, Table 1, the authors report the time constants of the generation of ferryl intermediates. How were derived the time constants? What is the equation used?
The transient rise and the subsequent decay, representing the formation and the decomposition of the ferryl states and the radical, were fitted by the two-step model
k1 k2
A ® B ® C
using equation B = A0 ((k1/(k2-k1))(e-k1t – e-k2t). Where k1 and k2 are corresponding rate constants and A0 is the initial concentration of the enzyme participating in the reaction. Then corresponding reaction time constants are equal to t = 1/k. This description is now included (lines 452 -456.
- The scheme on pag. 9 shows a P0form and the authors discuss (lines 227-234) on the transitions, between the different intermediates, that occur at different pH values. What are the features of the P0compound?
The formation of P0 intermediate, the metastable adduct of peroxide with heme a3, was suggested in the earlier works for the reaction of oxidized CcO with H2O2. The P0 intermediate is very likely singly or doubly ionized hydrogen peroxide bound to heme a3 forming metastable iron-peroxy adduct, Fe3+ -O-OH (or Fe3+-O-O-) analogous to compound 0 of peroxidases. This is now presented in Discussion (lines 258-261).
Minor points
- On page 12 line 396-397, the authors state: “The conditions in these spectral measurements were identical to those used in the ITC data collections.” However the experimental data are not reported in the ITC measurements paragraph.
This sentence was removed. All particular conditions of the measurements can be found in the Results section.
- Page 6, line 159 (Fig. 5A, F) and not (Fig. 5, F).
Corrected
- Page 6, line 174 (Fig. 5A, PMdotted line) and not (Fig. 5, PMdashed line).
corrected
- Page 7, line 188 (Fig. 5A, R•dashed line) and not (Fig. 5, R• dashed line).
Corrected
Reviewer 2
- In this work, the formation and the endogenous decay of both the ferryl iron of heme a3 and the radical in F• intermediate were examined by the combination of four experimental approaches, isothermal titration calorimetry, electron paramagnetic resonance, and electronic absorption spectroscopy together with the reduction of this form by the defined number of electrons.
As far as I know, there are some other characterization methods, such as time-resolved X-ray crystallography and single crystal microspectrophotometry, can study the formation and structure of intermediates. Please explain the reasons for choosing these four methods.
Selection of the employed methods for the present investigation was determined by the questions about the redox state of the catalytic center of the ferryl F• intermediate, the stability of the formed radical in this state and thermodynamics characterization of the formation of this form in the reaction of the oxidized CcO with hydrogen peroxide. Undoubtedly, these goals can be reached also by several other methods. However, we believe that the combination of ITC, EPR measurements together with the reduction of F• with ferrocyanide convincingly showed that the lifetime of radical is shorter relative to the ferryl iron of heme a3.
- In the process of research, are unexpected combination of other chemical groups during this reaction at a site distant fromfound?
Hopefully, we understand this question correctly.
If the catalytic center of CcO is blocked by cyanide and free cyanide is removed from the sample, there is no heat release and also the generation of the radical is missing after addition of peroxide to oxidized CcO. This indicates that there are no other groups, except the catalytic center, to which the observed phenomena (large heat release, formation of radical) can be assigned. The presented data showed that the origin of the observed process is the reaction of H2O2 with the heme a3-CuB center.
- In the conclusion paragraph, the author puts forward an interesting, important but speculative view- “The EPR data, the large enthalpy changes, and the reduction of the F• form by the defined amount of ferrocyanide showed that the migration of the radical from the catalytic center is also associated with its rapid quenching, very likely at the surface of the protein.” How to demonstrate this argument?
We are planning another study to support our conclusion on the radical migration during the generation and decay of the F•state by using spin traps. It can be expected that at large concentration of spin trap we should be able to trap and stabilize the radical at room temperature by EPR. This possibility is now mentioned in the revised manuscript (lines 385-386).

Reviewer 2 Report
Review comments
Various enzymes use semi-stable ferryl intermediates and free radicals during their catalytic cycle. In this article, the author investigated the radical in the peroxide-produced F-type ferryl form of bovine cytochrome c oxidase. The selected topic conforms to the positioning of the special issue. The research design is reasonable, and the research results can answer the proposed research questions to a certain extent.
Specific Questions:
1.In this work, the formation and the endogenous decay of both the ferryl iron of heme a3 and the radical in F• intermediate were examined by the combination of four experimental approaches, isothermal titration calorimetry, electron paramagnetic resonance, and electronic absorption spectroscopy together with the reduction of this form by the defined number of electrons.
As far as I know, there are some other characterization methods, such as time-resolved X-ray crystallography and single crystal microspectrophotometry, can study the formation and structure of intermediates. Please explain the reasons for choosing these four methods.
2.In the process of research, are unexpected combination of other chemical groups during this reaction at a site distant from found?
3.In the conclusion paragraph, the author puts forward an interesting, important but speculative view- “The EPR data, the large enthalpy changes, and the eduction of the F• form by the defined amount of ferrocyanide showed that the migration of the radical from the catalytic center is also associated with its rapid quenching, very likely at the surface of the protein.” How to demonstrate this argument?
Author Response
Reviewer 2
- In this work, the formation and the endogenous decay of both the ferryl iron of heme a3 and the radical in F• intermediate were examined by the combination of four experimental approaches, isothermal titration calorimetry, electron paramagnetic resonance, and electronic absorption spectroscopy together with the reduction of this form by the defined number of electrons.
As far as I know, there are some other characterization methods, such as time-resolved X-ray crystallography and single crystal microspectrophotometry, can study the formation and structure of intermediates. Please explain the reasons for choosing these four methods.
Selection of the employed methods for the present investigation was determined by the questions about the redox state of the catalytic center of the ferryl F• intermediate, the stability of the formed radical in this state and thermodynamics characterization of the formation of this form in the reaction of the oxidized CcO with hydrogen peroxide. Undoubtedly, these goals can be reached also by several other methods. However, we believe that the combination of ITC, EPR measurements together with the reduction of F• with ferrocyanide convincingly showed that the lifetime of radical is shorter relative to the ferryl iron of heme a3.
- In the process of research, are unexpected combination of other chemical groups during this reaction at a site distant fromfound?
Hopefully, we understand this question correctly.
If the catalytic center of CcO is blocked by cyanide and free cyanide is removed from the sample, there is no heat release and also the generation of the radical is missing after addition of peroxide to oxidized CcO. This indicates that there are no other groups, except the catalytic center, to which the observed phenomena (large heat release, formation of radical) can be assigned. The presented data showed that the origin of the observed process is the reaction of H2O2 with the heme a3-CuB center.
- In the conclusion paragraph, the author puts forward an interesting, important but speculative view- “The EPR data, the large enthalpy changes, and the reduction of the F• form by the defined amount of ferrocyanide showed that the migration of the radical from the catalytic center is also associated with its rapid quenching, very likely at the surface of the protein.” How to demonstrate this argument?
We are planning another study to support our conclusion on the radical migration during the generation and decay of the F•state by using spin traps. It can be expected that at large concentration of spin trap we should be able to trap and stabilize the radical at room temperature by EPR. This possibility is now mentioned in the revised manuscript (lines 385-386).
Round 2
Reviewer 1 Report
The authors have made satisfactory amendments to the manuscript in response to my previous comments.
I also note that great effort has been made to appease the other reviewer comments.
In my opinion this manuscript can be accepted for publication.
Reviewer 2 Report
In the revised version and the feedback letter, the author responded positively to the questions raised previously. I expect the author to produce more in-depth research results in the future.